# Resonant torsion magnetometry in anisotropic quantum materials

K.A. Modic[1], Maja D. Bachmann[1], B.J. Ramshaw [2], F. Arnold[1], K.R. Shirer[1], Amelia Estry[1], J.B. Betts[3], Nirmal J. Ghimire[3,5], E.D. Bauer [3], Marcus Schmidt[1], Michael Baenitz[1], E. Svanidze[1], Ross D. McDonald [3], Arkady Shekhter[4] & Philip J.W. Moll [1,6]

Unusual behavior in quantum materials commonly arises from their effective low-dimensional physics, reflecting the underlying anisotropy in the spin and charge degrees of freedom. Here we introduce the magnetotropic coefficient $k = \partial^2 F/\partial\theta^2$, the second derivative of the free energy $F$ with respect to the magnetic field orientation $\theta$ in the crystal. We show that the magnetotropic coefficient can be quantitatively determined from a shift in the resonant frequency of a commercially available atomic force microscopy cantilever under magnetic field. This detection method enables part per 100 million sensitivity and the ability to measure magnetic anisotropy in nanogram-scale samples, as demonstrated on the Weyl semimetal NbP. Measurement of the magnetotropic coefficient in the spin-liquid candidate $RuCl_3$ highlights its sensitivity to anisotropic phase transitions and allows a quantitative comparison to other thermodynamic coefficients via the Ehrenfest relations.

[1] Max-Planck-Institute for Chemical Physics of Solids, Noethnitzer Strasse 40, D-01187 Dresden, Germany. [2] Laboratory of Atomic and Solid State Physics, Cornell University, Ithaca, NY 14853, USA. [3] Los Alamos National Laboratory, Los Alamos, NM 87545, USA. [4] National High Magnetic Field Laboratory, Florida State University, Tallahassee, FL 32310, USA. [5] Present address: Argonne National Laboratory, Lemont, IL 60439, USA. [6] Present address: EPFL STI IMX-GE MXC 240, CH-1015 Lausanne, Switzerland. Correspondence and requests for materials should be addressed to K.A.M. (email: modic@cpfs.mpg.de) or to P.J.W.M. (email: philip.moll@epfl.ch)

Correlated quantum materials governed by strong electronic interactions commonly host a variety of competing and coexisting electronic phases, such as the copper- and iron-based high-$T_c$ superconductors where charge ordering, high-temperature superconductivity, and magnetism occur in close proximity[1]. Mapping the associated phase diagram is a critical first step to understanding their physics. These phases are commonly characterized by anisotropic behavior that reflects the microscopic anisotropy in the spin and charge degrees of freedom. Prominent examples include anisotropy in the magnetic susceptibility of the cuprates[2–4], the identification of hidden-order phases in $URu_2Si_2$ and $SmB_6$[5,6] and the electronic nematicity of the iron-based superconductors[7,8]. While anisotropy is an essential ingredient for the complex phases that emerge in quantum materials, its experimental signatures can be subtle.

An established and highly sensitive technique to directly probe small anisotropies in correlated metals and exotic magnets is torque magnetometry[9–11]. When a sample with an anisotropic magnetization $\mathbf{M}$ is placed in an external magnetic field $\mathbf{B}$, it experiences a torque $\tau = \mathbf{M} \times \mathbf{B}$. This torque can be measured with high accuracy by mounting a crystal onto a cantilever[12–17]. Because the overall susceptibility is small, we assume that the local field $B$ is approximately equal to the applied field $H$, and use $B$ throughout.

Both the magnetic torque $\tau = \partial F/\partial \theta$ and the magnetization $M = -\partial F/\partial B$ are first derivatives of the free energy $F$, and thus these thermodynamic potentials provide sensitive and essential information at phase transitions (Fig. 1a). Second derivatives of the free energy, however, such as the heat capacity $C = -T\partial^2 F/\partial T^2$, the magnetic susceptibility $\chi = -\partial^2 F/\partial B^2$, and the elastic moduli $c_{ijkl} = \partial^2 F/\partial \epsilon_{ij}\partial \epsilon_{kl}$ often provide more fundamental insights into a material. These quantities can be directly related to physical properties, such as the density of states, and are the essential quantities to formulate microscopic theories. Unlike first derivatives, they exhibit discontinuities at second-order phase transitions and their magnitudes can be related to one another through the Ehrenfest relations[18].

Here, we develop a technique to measure the curvature of a sample's free energy with respect to magnetic field orientation $k = \partial^2 F/\partial \theta^2$—the thermodynamic coefficient directly linked to magnetic anisotropy. We name this the magnetotropic coefficient as it describes rigidity with respect to rotation in a magnetic field. With the sample mounted onto a resonating cantilever, the magnetotropic response acts to reduce or enhance stiffness of the total system, leading to a shift in the resonant frequency (Fig. 1c). This is in contrast with the magnetic torque, which bends the cantilever to a new equilibrium position but does not lead to a frequency shift. The measured frequency is highly sensitive to the magnetic anisotropy of a sample, which is the basis of resonant torsion magnetometry[19,20]. We demonstrate the sensitivity of resonant torsion magnetometry and highlight the importance of its thermodynamic properties on two materials, one with charge- and the other with spin-dominated anisotropy. We measure quantum oscillations in the Weyl semimetal NbP[21,22] and the antiferromagnetic phase boundary of the spin-liquid candidate $RuCl_3$[23–29].

## Results

**Kinematics of resonant torsion.** To detect the magneotropic coefficient, we use the Akiyama Probe (A-Probe)—a self-oscillating

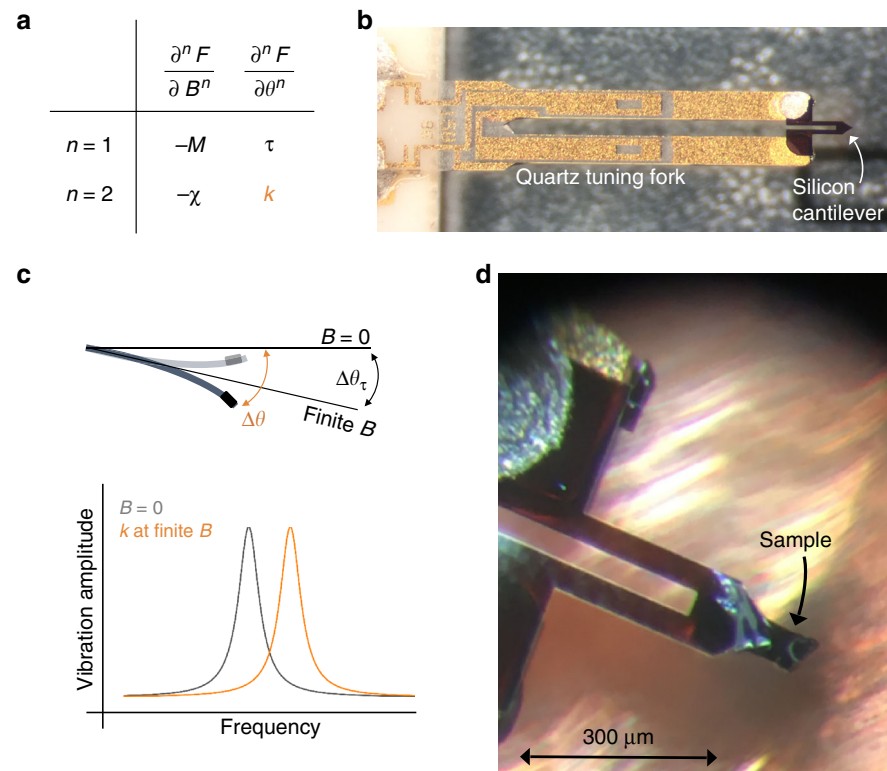

| | $\dfrac{\partial^n F}{\partial B^n}$ | $\dfrac{\partial^n F}{\partial \theta^n}$ |
|---|---|---|
| $n = 1$ | $-M$ | $\tau$ |
| $n = 2$ | $-\chi$ | $k$ |

**Fig. 1** Schematic overview of resonant torsion magnetometry. **a** First and second derivatives of the free energy with respect to the magnetic field $B$ and the field orientation $\theta$. **b** The quartz tuning fork of the Akiyama A-probe (http://www.akiyamaprobe.com) is electrically excited at the lowest-resonance mode of the silicon cantilever, producing a large out-of-plane motion at the tip of the cantilever. **c** Schematic representing the principle of measuring the magnetotropic coefficient $k$. In a magnetic field, the magnetic torque brings the lever to a new equilibrium position. The magnetic energy of the samples changes the effective stiffness of the lever, leading to a shift in the resonant frequency. **d** The silicon cantilever glued to each leg of the quartz tuning fork with a single crystal of $RuCl_3$ mounted at the tip with Bayer silicone grease

and self-sensing cantilever designed for scanning probe microscopy[30]. The A-probe is made of two separate resonators: a silicon U-shaped cantilever and a quartz tuning fork (see Fig. 1b and Methods)[30]. The benefits of repurposing the A-probe for resonant torsion magnetometry are threefold: the relatively large spring constant of the silicon cantilever (5 N m$^{-1}$[30]) allows us to extend ultrasensitive and dynamic cantilever magnetometry[31–36] to macroscopic sample sizes; placement of the sample on the silicon cantilever (rather than one leg of a quartz tuning fork) eliminates complications that arise from the center of mass motion of the tuning fork coupling to the resonance mode[37,38]; and electrical read-out of the A-probe eliminates the need for optical detection of the resonant frequency, making setup relatively straightforward and more robust compared to previous approaches.

To elucidate the physical distinction between the magnetic torque and the magnetotropic coefficient and to describe the measurement, we briefly review the energetics of the resonating sample. In the harmonic approximation, the energy of a cantilever with effective stiffness $K$, moment of inertia $I$ (see Methods) and an attached sample can be written as

$$E = \frac{I}{2}\left(\frac{d\Delta\theta}{dt}\right)^2 + \frac{K}{2}(\Delta\theta)^2 - \tau\Delta\theta + \frac{k}{2}(\Delta\theta)^2, \quad (1)$$

where $\Delta\theta$ describes the angle of rotation of the sample with respect to a fixed magnetic field. Note that this is opposite in sign to $\theta$ discussed elsewhere, which describes rotation of the magnetic field with respect to fixed crystal axes.

The first two terms describe the kinetic and potential energies of the bare cantilever and together determine the base oscillation frequency, $\omega_0^2 = K/I$. We parameterize the motion of the lever as it vibrates by an angle $\Delta\theta$ at the tip of the lever where the sample is mounted (Fig. 1c). The last two terms in Eq. (1) describe the anisotropic energy of the measured sample in the applied magnetic field. Both the torque and the magnetotropic coefficient appear as coefficients in a Taylor expansion of the free energy $F(\theta, B)$, and they manifest themselves in distinct physical responses of the sample. The torque shifts the equilibrium angle about which the lever oscillates to $\Delta\theta_\tau = \tau/(K + k)$ (Fig. 1c). The magnetotropic coefficient encodes the curvature of the free energy with respect to the rotation angle, and appears as a shift in the oscillation frequency $(\omega_0 + \Delta\omega)^2 = (K + k)/I$. For small frequency shifts, this can be expanded as

$$\frac{\Delta\omega(\theta, B)}{\omega_0} = \frac{k(\theta, B)}{2K}. \quad (2)$$

Therefore, $k$ can be directly determined by a measurement of the resonant frequency of the cantilever.

**Linear magnetic response**. In general, the functional form of the magnetization $M(B)$ can include terms other than those linear in magnetic field. These are common in magnetic materials, even at very low fields. Therefore, the torque $\tau = M(B) \times B$, and subsequently the magnetotropic coefficient $k = \partial\tau/\partial\theta$, can also carry a more complex form. We first focus on the simple case of the linear response regime ($M_i = \chi_{ij}B_j$), however, to illustrate the different behaviors of $\tau$ and $k$. Here, the free energy $F(\theta, B) = (1/4)(\chi_j - \chi_i)B^2\cos 2\theta$ requires the angle dependences of the torque $\tau \propto \sin 2\theta$ and the magnetotropic coefficient

$$k(\theta, B) = \left(\chi_i - \chi_j\right)B^2\cos 2\theta \quad (3)$$

to strongly differ. Here, $\theta$ is defined as the angle of rotation of magnetic field with respect to the $i$th crystal direction in a right-handed spherical coordinate system.

We observe the expected angle dependence for the magnetotropic coefficient in a resonant torsion measurement of RuCl$_3$ at low fields within the linear regime (Fig. 2). Importantly, the signal of resonant torsion is maximal for fields along the axes of symmetry, a disadvantageous field orientation for conventional torque measurements because the signal goes to zero. Even in the vicinity of these directions (gray lines in Fig. 2), the torque is subject to an undesirable interaction effect (see methods), which contributes minimally to the magnetotropic coefficient. While the magnetic torque and the magnetotropic coefficient are simply related to each other in the linear magnetic regime, we later capture the nonlinear response in RuCl$_3$ at higher magnetic fields and show that it conveys new information about the magnetic anisotropy, different from the magnetic torque.

**High sensitivity de Haas-van Alphen**. In order to demonstrate the sensitivity of the technique, we measure quantum oscillations in the Weyl semimetal NbP[21,22] up to 3 T (Fig. 3). This semimetal is non-magnetic, and its entire magnetic response at low fields is due to the weak Landau diamagnetism of the conduction electrons, as well as the Berry paramagnetism arising from its non-trivial topology[39,40]. With the magnetic field applied along the crystallographic $c$ axis, where the magnetic torque signal is zero, we can resolve quantum oscillations in fields well below 1 T. The quantum oscillation frequencies for this field orientation agree with those reported in the literature[21,22]. With a characteristic response bandwidth of 1 Hz, the smallest detectable frequency shift is $\Delta f/f = 6 \times 10^{-9} = \Delta k/K$, where $K$ is the effective bending stiffness of the lever (see Methods). With $K = 180$ nJ rad$^{-2}$, the smallest detectable magnetotropic coefficient is $\Delta k = 1.1 \times 10^{-15}$ J rad$^{-2}$, equivalent to $1.2 \times 10^8$ $\mu_B$ at 1 T. This can be used to estimate the required mass of a metallic crystal that can be investigated with resonant torsion magnetometry. Even in only weakly anisotropic metals (1% anisotropy), which would contribute 0.01 $\mu_B$ per formula unit, only $10^{12}$ formula units are needed to resolve a signal at the demonstrated sensitivity. For a 3 Å unit cell size, this corresponds to a 3 $\mu$m$^3$ sample size or a sample weight of 0.1 ng for a sample density of 5 g cm$^{-3}$. Resonant torsion magnetometry is thus ideally suited to investigate

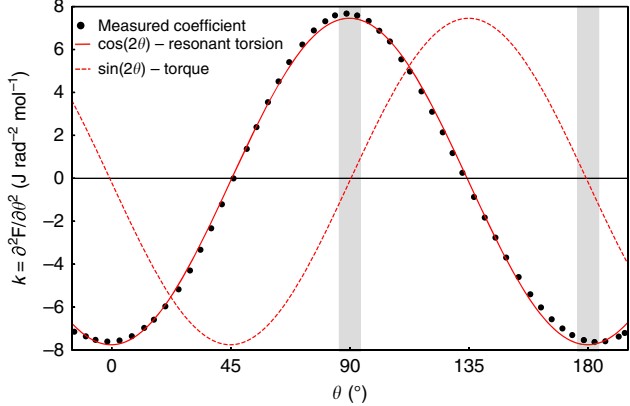

**Fig. 2** Angle dependence of the magnetotropic coefficient. The magnetotropic coefficient $k$, proportional to the shift in frequency, of RuCl$_3$ at $T = 16$ K and $B = 5$ T (black points). The expected angle dependences of the magnetotropic coefficient (solid red line) and the magnetic torque (dashed red line) in the linear response regime $M_i = \chi_{ij}B_j$ are overlaying the data. $\theta = 0°$ and $\theta = 90°$ correspond to magnetic field applied perpendicular and parallel to the honeycomb planes, respectively. In the linear response regime, the principal magnetic axes (gray bands) have a maximal response in the magnetotropic coefficient, and these directions coincide with the zeros of the torque signal

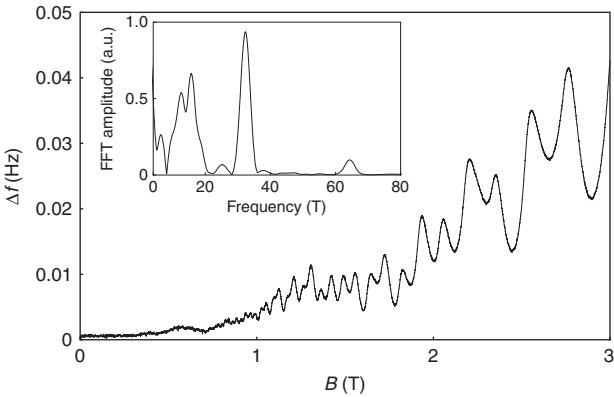

**Fig. 3** Quantum oscillations in the Weyl semimetal NbP. At a temperature of 1.2 K, de Haas–van Alphen oscillations of NbP up to 3 T as detected with resonant torsion magnetometry. The inset shows the Fourier spectra. Magnetic field is applied along the crystallographic c axis. The measured noise with the PLL bandwidth of 1 Hz in zero magnetic field is roughly 300 µHz

anisotropy when only the smallest samples exist in single crystal form.

**Thermodynamics and phase transitions.** The magnetotropic coefficient can provide valuable insight into the thermodynamics of phase transitions via the Ehrenfest relations. $k$ can be more formally defined as a member of a matrix of second derivatives (thermodynamic coefficients) of the free energy when temperature $T$, volume $V$, magnetic field $B$, and magnetic field orientation $\theta$ are independent variables. The relation of $k$ to other thermodynamic coefficients is directly apparent from the behavior of the thermodynamic potential in the $T$, $V$, $B$, and $\theta$ variables

$$dF = -SdT - PdV - MdB + \tau d\theta. \quad (4)$$

We can derive the Ehrenfest relation that relates a discontinuous jump in the resonant torsion to other thermodynamic coefficients. If we assume that $T_c(\theta)_{V,B}$ is the boundary of a second-order phase transition induced by the magnetic field angle measured at a fixed volume $V$ and magnetic field $B$, then continuity of all first derivatives ($S$, $P$, $M$, $\tau$) across such a boundary, $\Delta S = 0$ and $\Delta \tau = 0$, requires that discontinuous jumps in the three thermodynamic coefficients $C$, $(\partial S/\partial \theta) = -(\partial \tau/\partial T)$, and $k$ are all related to each other:

$$\frac{\Delta C}{T_c}dT^* + \Delta(\partial S/\partial \theta)d\theta^* = 0$$
$$\Delta(\partial \tau/\partial T)dT^* + \Delta k d\theta^* = 0. \quad (5)$$

Here, $\Delta X$ indicates the jump of $X$ across the phase boundary and $dT^*$ and $d\theta^*$ are short segments along the phase boundary in the $T - \theta$ phase plane, such that $dT^*/d\theta^* = (\partial T_c/\partial \theta)_B$. The Ehrenfest relation connecting the jump in the magnetotropic coefficient $\Delta k$ and the jump in the heat capacity $\Delta C$ is

$$\Delta k = -\frac{\Delta C}{T_c}(\partial T_c/\partial \theta)_B^2, \quad (6)$$

where the derivative is to be taken along the phase boundary at fixed magnetic field. Similarly, Ehrenfest relations between the jumps in $k$, $\chi$, and $C$ give $\Delta k = -\Delta \chi (\partial B_c/\partial \theta)_T^2$ and $\Delta \chi = (\Delta C/T_c)(\partial T_c/\partial B)_\theta^2$, where the derivatives in the two relations must be taken along the phase boundary at fixed temperature and at a fixed field orientation, respectively.

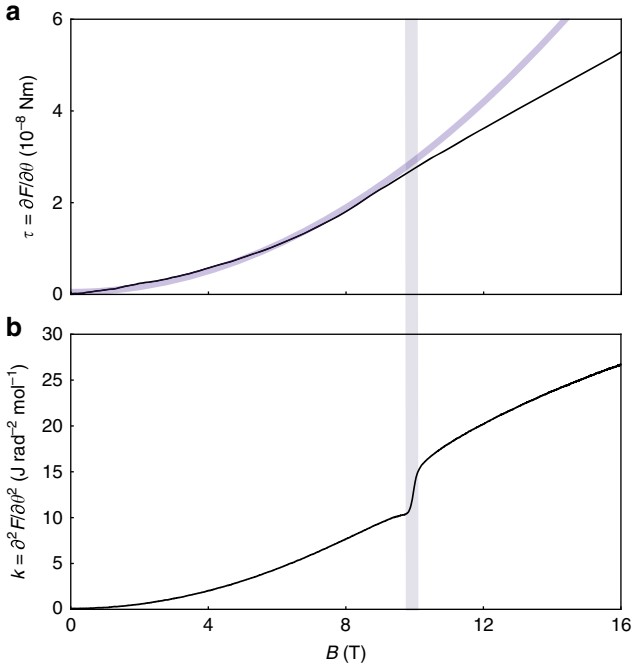

**Fig. 4** Crossing the antiferromagnetic phase boundary in RuCl$_3$. Comparison of **a** the magnetic torque $\tau$ inferred from the bending amplitude of a Seiko cantilever and **b** the magnetotropic coefficient $k$ measured with resonant torsion magnetometry of RuCl$_3$ at $T = 1.3$ K. Magnetic field is applied at an angle ~10° from the honeycomb planes toward the crystallographic c axis. Both measurements probe the magnetic anisotropy between the in-plane and out-of-plane field orientations (i.e., $\alpha = \pm(\chi_\parallel - \chi_\perp)$, where $\chi_\parallel$ and $\chi_\perp$ are with respect to the honeycomb planes). The gray band highlights the field position of the suppression of long-range antiferromagnetic order. At the phase transition, the torque shows a deviation from the low-field quadratic response (blue curve), while the magnetotropic coefficient features a sharp jump

In order to verify experimentally these thermodynamic relations, we refer again to RuCl$_3$, an effective spin-1/2 quantum magnet that orders antiferromagnetically at $T_N = 7$ K[26]. Below this temperature, long-range order can be suppressed with a magnetic field of ~8 T for fields applied within the honeycomb planes[28,29], with recent evidence suggesting a spin-liquid state at higher magnetic fields[23]. We measure RuCl$_3$ at $T = 1.3$ K—well within the antiferromagnetically ordered state[27,29]—to observe the evolution of the magnetic torque and the resonant torsion as we cross the second-order phase boundary with increasing magnetic field (Fig. 4). The torque (Fig. 4a) is inferred from the piezoresistively detected bending amplitude of a Seiko Instruments cantilever[14]. With small fields at an angle ~10° away from the honeycomb planes, both $\tau$ and $k$ respond quadratically to the applied magnetic field. For this field orientation, we observe the suppression of long-range order at ~9 T. Across the phase boundary (gray line in Fig. 4), $\tau$ shows a break in slope crossing over to linear behavior at higher magnetic fields, whereas $k$ experiences a discontinuous jump. Akin to the advantages of techniques sensing the magnetic susceptibility compared to magnetization, detecting $k$ offers a more appropriate means for identifying magnetic phase transitions.

The experimentally observed jump $\Delta k \approx 6$ J rad$^{-2}$ mol$^{-1}$ in this configuration (Fig. 4b) can be directly compared to heat capacity measurements under magnetic field. $(\partial T_c/\partial \theta)_B$ can be estimated from the angle dependence of the resonant torsion of RuCl$_3$ at fixed temperatures. One such scan at $T = 1.3$ K and $B = 17.5$ T shows a pronounced anomaly at the phase boundary of the long-

range ordered state (blue vertical line in Fig. 5). Entry into the ordered state is marked by a jump down at the phase boundary, as required by Eq. (6). Similar measurements at various fixed magnetic fields allow to map out the phase boundary of the antiferromagnetically ordered state (Fig. 5b). The derivative $(\partial T_c/\partial\theta)_B = (\partial T_c/\partial B)_\theta\ (\partial B_c/\partial\theta)_T$ at $T = 1.3$ K, B = 10 T, and $\theta = 102°$ can be estimated as $(\partial B_c/\partial\theta)_T \approx 2.8$ T rad$^{-1}$ and $(\partial T_c/\partial B)_\theta \approx 25$ K

T$^{-1}$. The heat capacity jump at the antiferromagnetic transition at $T = 1.3$ K has been reported as $\Delta C/T_c \sim 1.7$ mJ mol$^{-1}$ K$^{-2}$[28]. Thus, the right-hand side of Eq. (6) gives ~8 J rad$^{-2}$ mol$^{-1}$, in agreement with the size of the measured jump $\Delta k$ of 6 J rad$^{-2}$ mol$^{-1}$ found above. This quantitative agreement is remarkable, especially given the uncertainties of the derivatives due to the complex shape of the phase boundary.

## Discussion

The magnetotropic coefficient provides valuable thermodynamic information and complements the magnetic torque. The direct measurement of $k$ highlights second-order phase transitions by discontinuous jumps that can be related to anomalies in other thermodynamic measurements, such as the heat capacity. Resonant torsion allows direct access to the magnetic anisotropy when the magnetic field is aligned along the principal magnetic axes—a blindspot for conventional torque magnetometry. Finally, the ability to measure shifts in the resonant frequency of lever vibrations much more precisely than the amplitude of lever deflections results in better than part per 100 million sensitivity and the opportunity to measure sub-nanogram samples.

## Methods

**Characteristics of the cantilever**. The A-probe, originally designed for atomic force microscopy (AFM), consists of a piezoelectric quartz tuning fork, with a silicon cantilever glued to the ends (Fig. 6). Electrical grounding of the silicon tip can be made via the blob of silver epoxy on one contact (Fig. 6a). Each tuning fork leg is 2400 μm long and the cross-sectional area is $124 \times 214$ μm$^2$ (Fig. 6b). The silicon cantilever is $l = 310$ μm long, $w = 30$ μm wide (Fig. 6a), and 3.7 μm thick[30]. We estimate the mass of the lever to be 100 ng, much smaller than the mass of the tuning fork.

In Eq. (1) in the main text, the potential and kinetic energy of the vibrating cantilever is described with a bending stiffness $K$ and a moment of inertia $I$. The bending stiffness $K$ is the coefficient of elastic energy stored in the silicon cantilever when the tip of the lever is bent by an angle $\Delta\theta$. The stored elastic energy, therefore, depends not only on the geometry (width, length, etc.) of the lever, but also on the shape of the resonance mode.

In the main text, we calibrate the shift in frequency using known values of the anisotropic susceptibility in the linear response regime, and its connection to the magnetotropic coefficient $k = (\chi_c - \chi_a)B^2 \cos(2\theta)$. Alternatively, we can estimate the magnitude of the magnetotropic coefficient $k$ for RuCl$_3$ using Eq. (2) in the main text. First, we need to take into account the shape of the bending mode of the cantilever to determine the bending stiffness $K$.

The bending shape of the silicon cantilever is described by $\zeta(z, t)$, where $z$ is the distance along the lever from the point of attachment and $\zeta$ is the vertical displacement of the lever[41]. At resonance, the motion of the lever is described by $\zeta(z, t) = \zeta^{(n)}(z)\sin(2\pi f^{(n)}t)$, where $\zeta^{(n)}(z)$ is the shape of the $n$-th mode and $f^{(n)}$ is its frequency. The shape of the lever is found from elastic equations derived from an

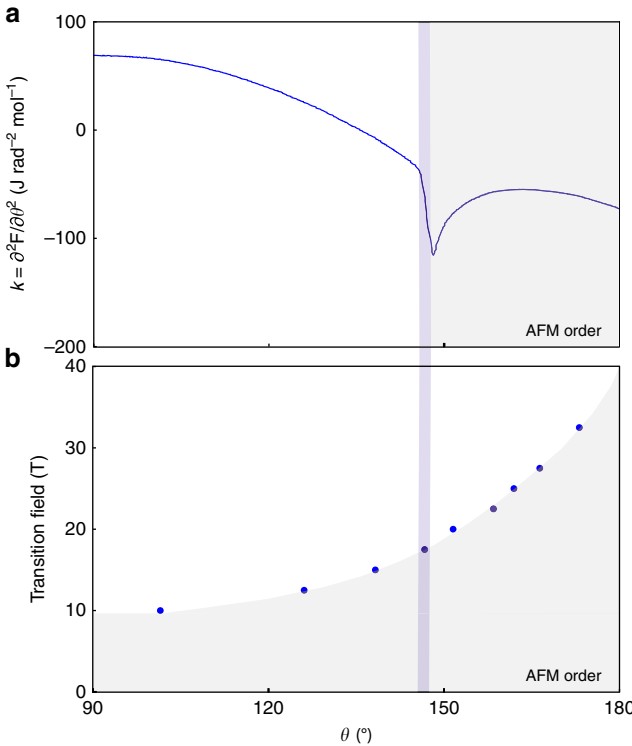

**Fig. 5** Anisotropic phase boundary in RuCl$_3$. **a** The magnetotropic coefficient $k$ of RuCl$_3$ at $T = 1.3$ K and $B = 17.5$ T as magnetic field is rotated from the honeycomb planes ($\theta = 90°$) to the perpendicular orientation. Entry into the long-range antiferromagnetically ordered state (gray region) is marked by a sharp jump down in the magnetotropic coefficient (vertical blue line). **b** Field orientation scans at various fixed magnetic fields (similar to **a**) allow a complete mapping of the antiferromagnetic phase boundary

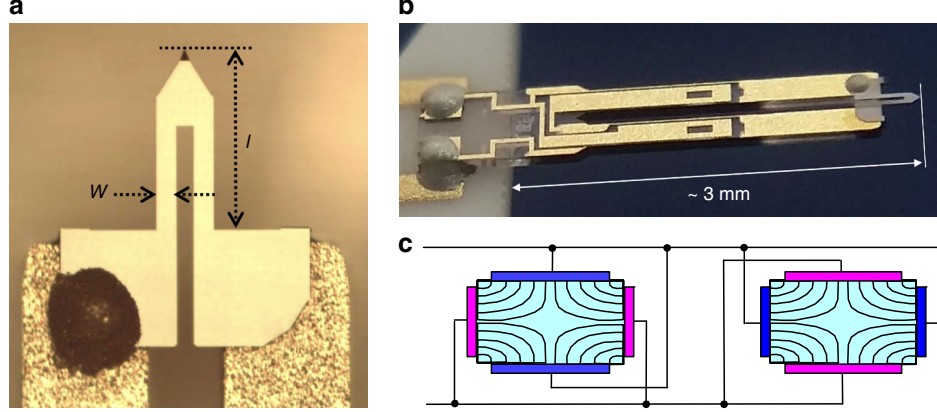

**Fig. 6** NANOSENSORS™ Akiyama-probe. **a** Photograph of the silicon cantilever highlights the attachment to the quartz tuning fork (reproduced from http://www.akiyamaprobe.com). Glue lies between the silicon and ends of the tuning fork legs. The left contact is covered in silver epoxy to ground the tip of the cantilever in AFM[30] (http://www.akiyamaprobe.com). **b** Photograph of the gold-covered quartz tuning fork and the silicon cantilever (reproduced from http://www.akiyamaprobe.com). **c** Cross-sectional area of the quartz tuning fork, where the drive and pickup gold contacts along the length of the lever are represented by blue and pink, respectively. The quadrupolar electric field induces motion in the tuning fork

energy functional $E = (1/2)\rho A \int_{dz}(d\zeta(z,t)/dt)^2 + (1/2)YI_c\int_{dz}(d^2\zeta(z,t)/dz^2)^2$, where the first term is the kinetic energy ($\rho$ is the density and $A$ is the cross-sectional area) and the second term is the potential energy ($Y$ is Young's modulus and $I_c$ is the moment of inertia of the lever's cross-section with respect to its center of mass[41]). The second derivative $d^2\zeta(z,t)/dz^2$ in the elastic energy represents the inverse radius of local curvature. Evaluating the second term for the fundamental vibration mode $\zeta^{(0)}$ (normalized as $d\zeta(z)/dzz = $ tip of the lever $= \Delta\theta(t)$) gives $K^{(0)} = 1.63(YI_c/L)$, which for the silicon lever evaluates to $K^{(0)} = 180$ nJ rad$^{-2}$. Similarly, evaluating the first term results in the moment of inertia for the same mode as $I^{(0)} = 0.13\rho AL^3 = 0.19 \times 10^{-17}$ J Hz$^{-2}$.

For RuCl$_3$, calibration of the magnetotropic coefficient using the linear response regime $k = (\chi_c - \chi_a)B^2 \cos(2\theta)$ yields 1 Hz $= 0.321$ J rad$^{-2}$ mol$^{-1}$. From the dimensions, we estimate the sample mass to be ~20 ng, which corresponds to 25 picomol per unit cell of RuCl$_3$ (where the unit cell contains 4 formula units). This gives 8 pJ rad$^{-2}$ for the magnetotropic coefficient $k$ of the sample. Using $K^{(0)} = 180$ nJ rad$^{-2}$ in Eq. (2), the right-hand side evaluates to $2.2 \times 10^{-5}$, which is close to the expectation on the left-hand side for a 1 Hz shift in frequency.

The two gold contacts wrap around each leg to create quadrupolar electric field lines (Fig. 6c) when a voltage is applied. In resonant torsion magnetometry, the gold contacts are effectively used as a driver and a pickup. Applying a voltage induces motion due to the piezoelectric properties of the quartz. The large mechanical motion of the silicon cantilever on resonance drives a piezoelectric current that is detected in our measurement. In addition to this piezoelectric current, a background current is present at all frequencies due to the parasitic capacitance of the wires connecting the cantilever in the cryostat to the room temperature electronics.

The pickup voltage near a resonance has a standard Lorentzian shape, $V = V_{BG} + A/[\omega - \omega_0 + i\Gamma/2]$, where $V_{BG}$ is a background voltage due to the background current discussed above. $\omega_0$ and $\Gamma$ are the resonant frequency and linewidth, respectively (Fig. 7b). In the complex plane, plotting the imaginary versus real parts of the Lorentzian traces a circle. Any anharmonic deviation from Eq. (1) of the main text leads to a distortion of this circle. We find that the response of the Akiyama probe deviates from a circle, signaling the nonlinear response regime of the lever, when driven with oscillating voltages in excess of 100 mV ($V_{osc}$ in Fig. 8). We also checked optically that a driving voltage of 1 V leads to a displacement of the lever of about 2 degrees. Our typical driving voltage of 10 mV therefore is accompanied by much smaller angular displacements, ensuring the validity of Eq. (1).

**Tracking the resonant frequency.** Our measurement requires a method for following the resonant frequency as a function of temperature, magnetic field, and magnetic field orientation. Typically, this can be achieved with a phase-locked loop (PLL) controlled lock-in amplifier. We used the readily available (PLL/PID) option of the Zurich Instruments mid-frequency lock-in (MFLI) amplifier for fast and sensitive response to shifts in the resonant frequency as a function of magnetic field only. As a function of temperature and in some situations, such as measurement across a sharp phase transition, we use our custom software-implemented PLL. This allows us to obtain additional information at frequencies near the resonance, but it significantly slows down the measurement speed. Below we explain a critical limitation with the hardware-implemented PLL due to the background impedance detected in our measurement. We discuss how we overcome this with a capacitance compensation circuit (Fig. 8) when using the hardware-implemented PLL, and how this is resolved in software when frequency scans through the resonance are necessary.

For robust tracking, the hardware PLL requires a large phase swing across the resonant frequency. The largest phase swing of 360° is obtained when the reference point for the phase is inside of the Lorentzian circle ($Z_1$ in Fig. 7). The PLL implementation of the Zurich MFLI calculates the phase with respect to zero voltage only. Therefore, to achieve a large phase swing across the resonance, the circle of the Lorentzian must be in the vicinity of zero voltage in the complex plane. In the actual measurement, this circle is shifted away from zero (blue curve in Fig. 7) due to the finite background voltage arising from parallel capacitance in the cables (outside and inside of the measurement probe). The circle diameter, which is proportional to the amplitude of the drive voltage and inversely proportional to the linewidth of the resonance, is typically 10–100 µV for a resonance in air with a $q$-factor of ~2000. In vacuum, the $q$-factor is in excess of 10,000 and the circle diameter increases above 100 µV, leading to a larger phase swing for the same given background voltage.

There are three different situations for the phase swing on resonance. First, the zero voltage is inside the circle ($Z_1$ in Fig. 7) and the phase swing is monotonic as a function of frequency. This is most preferred for successful tracking. Second, the zero voltage lies outside of the circle and the phase swing is non-monotonic (the phase returns to the same value on both sides of the resonance). When the angle range (visible from $Z_2$ in Fig. 7) of the Lorentzian circle is not too small (roughly >30°), the PLL implementation of the Zurich MFLI can reliably still lock in to the resonance, but is sometimes an unstable situation. For example, if the system goes through a phase transition and the resonance becomes much broader, it may be lost. Third, the border of the circle crosses the zero voltage at the tail of the resonance (which never happens). This corresponds to no background voltage $V = A/[\omega - \omega_0 + i\Gamma/2]$ and a 180° phase swing. In the special case for the border of the circle near the zero voltage, the phase first swings slowly 180°, followed by an

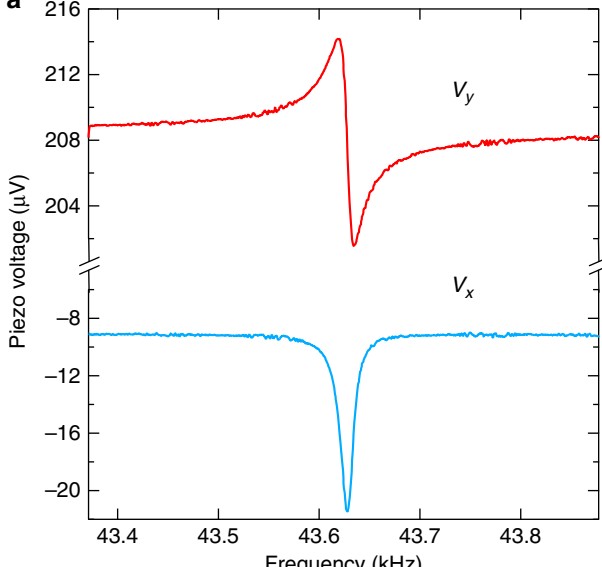

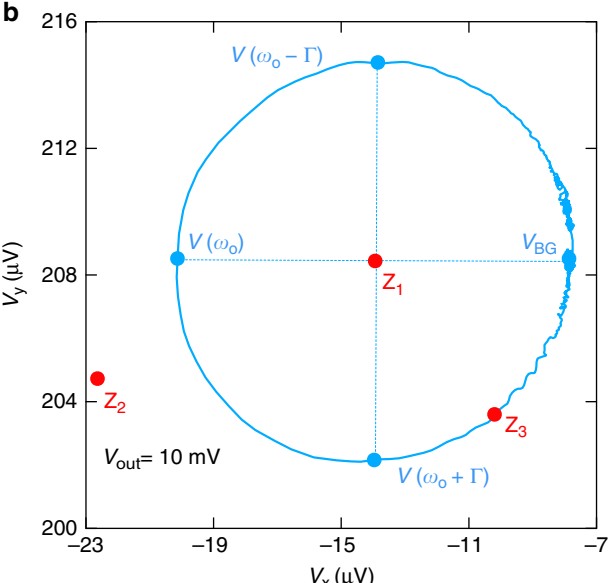

**Fig. 7** Cantilever resonance. **a** Frequency scans through the normal mode resonance at room temperature and in air. The in-phase and out-of-phase voltages are shown in blue and red, respectively. **b** The Lorentzian at resonance can be observed as a circle when plotted in the complex plane. Off resonance, the background voltage $V_{BG}$ is measured. Approaching the resonance corresponds to tracing the circle in a counter-clockwise direction starting from $V_{BG}$. The resonant frequency is observed at $V(\omega_o)$. The positions $Z_1$, $Z_2$, and $Z_3$ in the complex plane are different possible positions of the reference point (zero voltage) from which the phase can be measured with the PLL. Depending upon the drive voltage and the capacitance compensation, the circle can be shifted to encompass, lie outside, or lie on the border of the phase reference. The effect this has on the phase swing at resonance is discussed in the text. Distortions of the circle, which are not observed in our measurements with drive voltages below 100 mV, are an indication of the nonlinear response regime of the lever

abrupt additional 180° swing either up or down depending upon the position of the resonance (either slightly inside or slightly outside of the circle)—this is usually the situation that arises with use of a capacitance compensation circuit (Fig. 8).

In order to improve the phase swing when the background voltage is large (or the circle diameter is not large enough), we incorporate a modified capacitance compensation circuit (Fig. 8) based on the one recommended by

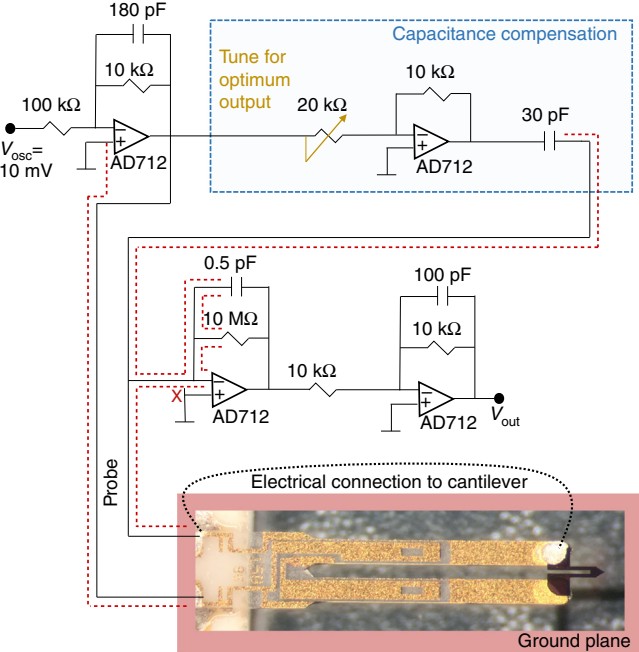

**Fig. 8** Detection circuit. Schematic diagram of the electrical circuit including necessary capacitance compensation elements, amplification, and placement of the A-Probe[30] in the circuit. The output from the capacitance compensation line is inverted and summed with the measured response of the cantilever to detect only the piezoelectric current due to the lever motion at resonance. To reduce noise, the braided shield of a low-capacitance twisted pair directly connects to the positive input of the operational amplifiers AD712, and to a ground plane directly below the vibrating lever. Our design is a modification based on that developed by NANOSENSORS™[30,45] (http://www.akiyamaprobe.com)

NANOSENSORS™[30]. The first stage following the input (upper left) acts as a buffer. An effective negative capacitance is then added in parallel with that of the measurement cables and the A-probe. The piezoelectric current, as well as the background current, is then detected at the current–voltage converter (marked by the red cross) and then amplified. At zero magnetic field, the background capacitance is nulled with the potentiometer until the phase shift on resonance is maximized, allowing successful tracking with the hardware PLL. Because the background capacitance changes as a function of temperature, we use our custom program to adaptively follow the resonance with temperature. Our software-implemented PLL uses feedback from previous frequency scans to measure phase from arbitrary points in the complex voltage plane[42,43].

**Regarding the linewidth**. These measurements also allow us to measure the linewidth evolution with temperature and magnetic field, which provides relaxation time information about anisotropic fluctuations. Excluding any experimental artifacts, the linewidth is directly determined by the energy dissipation per oscillation cycle[44], which in this measurement will be associated with magnetically anisotropic fluctuations. Here we discuss some experimental factors that are unrelated to the physics in the sample. This is especially important because causality (expressed via Kramers–Kronig relations) requires that changes in the linewidth are accompanied by related changes in the frequency[44]; when the linewidth decreases by an amount $\Delta\Gamma$, the frequency increases by a comparable amount. For example, bringing the lever into vacuum at room temperature removes the dissipation associated with air friction around the lever, which typically decreases the linewidth by about 20 Hz. This is accompanied by a 20 Hz increase in frequency that is observed in our measurement. This is of concern only when the frequency shifts are smaller than the linewidth. In our RuCl$_3$ measurements, the linewidth in vacuum at cryogenic temperatures is a couple of Hz and the frequency shifts observed in nanogram-sized samples under magnetic field/temperature are typically 100–1000s of Hz. Normally, we use 10 mbar of helium-4 exchange gas, which allows for a $q$-factor that can be as much as 30,000 at cryogenic temperatures. We have also observed unexpected frequency shifts with temperature that we believe result from partial covering of the lever with grease (used for sample attachment). The grease freezes below about 200 K, effectively increasing the bending stiffness of the lever by as much as 1%.

**Magnetotropic coefficient in the linear magnetic regime**. We now refer to the linear response regime ($M_i = \chi_{ij}B_j$) to compare the magnitude of the frequency shift $\Delta\omega/\omega_0$ and the average deflection angle $\Delta\theta_\tau$. The free energy of the sample as it rotates in a magnetic field is

$$F(\theta, B) = \frac{1}{4}\alpha B^2 \cos 2\theta, \qquad (7)$$

where $\alpha = \chi_j - \chi_i$ is the anisotropic susceptibility restricted to the plane of vibration. In the linear regime, $\tau = 2F(\theta, B)\tan 2\theta$ and $k = 4F(\theta, B)$ so that both the frequency shift and deflection angle can be written as a function of the free energy

$$\Delta\theta_\tau = 4\frac{F(\theta, B)}{K}\tan 2\theta, \quad \frac{\Delta\omega}{\omega_0} = 2\frac{F(\theta, B)}{K}. \qquad (8)$$

This shows that relative frequency shift in the linear regime is of the same order of magnitude as the bending angle:

$$\frac{\Delta\omega}{\omega_0} = \frac{\Delta\theta_\tau \cot 2\theta}{2}. \qquad (9)$$

Although the shift in frequency is accompanied with (and is proportional to) the average bending angle of the lever due to the magnetic torque, the former is not caused by the latter. These are two independent phenomena and they will start to affect each other if anharmonic effects become important—both in the response of the lever due to a large $\Delta\theta_\tau$ or in the sample due to a nonlinear magnetic response. In particular, for nonlinear magnetization, $\Delta\omega$ and $\Delta\theta_\tau$ are not related in a simple way.

**Comparison with conventional torque magnetometry**. As stated in the main text, resonant torsion magnetometry allows us to probe magnetic anisotropy in highly anisotropic systems along the crystallographic directions—often the main goal of an experiment. In order to detect the intrinsic magnetic anisotropy with conventional torque magnetometry, one must always apply field off of these directions to avoid interaction effects. This arises because the magnetic torque signal goes to zero near the principal directions (gray line in Fig. 2). Here (just like at all field orientation angles), the lever reorients in the magnetic field due to the torque. The sample then experiences a new torque due to the reorientation, but the change in these two torques is of the order of the total torque signal size. This behavior leads to a nonlinear response near the crystallographic axes[10], which can be avoided by measuring the magnetotropic coefficient instead of torque.

We find that use of the A-probe allows us to overcome several other systematic challenges associated with torque that is inferred from the angular deflection of a piezoresistive cantilever[12,14]. These include the additional deflection due to the force from magnetic field gradients, a magnetoresistive contribution, and an asymmetry in the response of the amplitude-detection levers. In our resonant torsion measurements, we are always operating within the linear response regime of the lever. The shift in the resonant frequency of the vibrating cantilever is capacitively inferred from an impedance measurement, alleviating the effects of magnetoresistance. Furthermore, shifts in the frequency of lever vibrations produced by spatial inhomogeneities of the magnetic field are much smaller than the corresponding shift in the average deflection angle of the lever, as demonstrated herein. In addition, our experimental setup allows for easy rotation with respect to the applied magnetic field, even within the confines of a pulsed field magnet. The high eigenfrequency of the cantilever and the high $q$-factor allow for fast response and high sensitivity on the ~10 ms timescale of pulsed magnetic fields. In this environment, where magnetoresistance can dominate in piezoresistive torque magnetometry, resonant torsion can be a marked advancement.

## Data availability

All relevant data are available upon request from the authors.

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

## Acknowledgements

The authors would like to thank the electronics workshop at the Max Planck Institute for Chemical Physics of Solids, particularly Dominic Hibsch, Wolfgang Geyer, and Torsten Breitenborn. We also thank Terunobu Akiyama for helpful discussions. Synthesis and characterization of the NbP single crystals was performed at Los Alamos National Laboratory under the auspices of the US Department of Energy, Office of Basic Energy Sciences, Division of Materials Sciences and Engineering. The portion of this work completed at the National High Magnetic Field Laboratory is supported through the National Science Foundation Cooperative Agreement numbers DMR-1157490 and DMR-1644779, The United States Department of Energy, and the State of Florida. M.D.B. acknowledges studentship funding from EPSRC under grant no. EP/I007002/1. R.D.M. acknowledges support from LANL LDRD-DR 20160085 topology and strong correlations. K.A.M. and P.J.W.M. acknowledge support of the Max Planck Society.

## Author contributions

K.A.M., R.D.M., A.S. and P.J.W.M. conceived of the experiment; A.E., N.J.G., E.D.B., M. S. and M.B. prepared and characterized the samples; K.A.M., M.D.B., K.R.S., J.B.B., E.S., R.D.M. and A.S. performed the experiments. K.A.M., B.J.R., F.A., E.S., R.D.M., A.S. and P.J.W.M. analyzed the data. K.A.M., B.J.R., A.S. and P.J.W.M. wrote the manuscript with input from all co-authors.

## Additional information

**Competing interests:** The authors declare no competing interests.

