## [Peer Review File · Nature Communications]

Reviewers' comments:

Reviewer #1 (Remarks to the Author):

The authors introduce a new thermodynamic quantity the magnetic torque susceptibility or magnetotropic coefficient as a probe for anisotropic materials. They present a theoretical framework, a method for measuring the magnetotropic coefficient, as well as examples of measurements on two selected compounds, NbP and RuCl₃.

To the best of my knowledge, the authors indeed establish a new method for the investigation of (anisotropic) magnetic materials. They successfully outline the major framework of the method. However, I have some hesitations with the manuscript.

1) The manuscript describes a new method to study materials. Traditionally, such manuscripts are published in dedicated journals like Review of Scientific Instruments. The editors should judge whether Nature Communications is a suitable platform. I expect that the present manuscript can make a substantial impact to the field and can stimulate new research and insight. The manuscript focuses on the presentation of the new thermodynamic quantity and method and does not present substantial insight into the physics of a specific material.

2) Further details are needed in several parts of the manuscript.

a. Equation (1) includes a Taylor expansion in free energy $F(\theta, H)$. The authors should comment on range of validity including quantitative estimates.

b. The authors mention an equilibrium position $\theta_{\tau} = \tau/K$. I believe, this should be $\theta_{\tau} = \tau/(K+k)$.

c. The authors briefly discuss and compare torque and magnetotropic coefficient in linear regime. They clearly highlight the benefits of magnetotropic coefficient but remain vague as to limitations arising from non-linear magnetisation. More details and quantitative estimates for non-linear effects on the magnetotropic coefficient are desirable. In fact, the wording of the sentences discussing non-linear effects should be improved.

d. The experimental details in the methods sections should be presented more clearly.

E.g.

i. Not clear if the drive voltage is oscillating

ii. Highlight current-to-voltage converter in sketch

iii. Describe role of electrical connection to cantilever (dashed line on picture in Fig6).

iv. The authors indicate that "the electrical resonance of the parallel LC circuit" is measured. This statement should be clarified, inductive and capacitive components should be highlighted in description and sketch.

I assume that the same method as in Ref. 32 is used. If so, this should be clearly referenced.

v. The magnetotropic coefficient is computed from the shift in frequency using equation (2). This requires knowledge of the effective stiffness. The authors provide an equation for the effective stiffness in the methods section $K_0 = 1.63(E I_y / L)$. The authors should provide the numerical value used for K including an error analysis (e.g. stemming from the uncertainty in dimensions and

density). In addition, I suggest to clarify whether I_y is assumed independent of position as it was introduced as $I_y(z)$.

In addition, the integral for the energy functional contains $E(z)$. The authors should highlight how the functional is evaluated.

vi. An additional frequency shift might occur due to a change in dissipation. The authors should discuss whether this can be neglected.

In summary, the manuscript presents a valuable step forwards for investigating anisotropic quantum materials as claimed in the title. If the authors can clarify the technical issues raised I am happy to recommend publication and expect that the manuscript will be of broad interest to experimental physicists.

Reviewer #2 (Remarks to the Author):

Dear Editor,

the manuscript by Modic and coworkers describes a technique to measure magnetic phase transitions using a method derived from torque magnetometry. The authors introduce a „magnetotropic“ term into the energy balance of the cantilever and show that they can measure this term by analysis of the resonance frequency. This is of potential great interest when exploring phase boundaries / transitions.

Although I find this method new and interesting I still have various questions regarding the technique, as well as the separation between the torque and the magnetotropic component, before I can recommend the manuscript for publication.

In detail, I encourage to address the following points

- 1) As shown in Figure 1d, the authors glued a massive object to the cantilever. As shown in the microscope image, the sample seems to exceed the dimensions of the cantilever, warranting the question whether the description of Eq1 holds. Please comment also on the quality factor of the cantilever.
- 2) The magnetic field orientation with respect to the crystalline axis of the sample and the cantilever is a bit unclear. A figure / inset would help here.
- 3) As the authors state, also torque magnetometry can be measured via the frequency shift of the mechanical system. Therefore, I encourage the authors to explain in more detail how they separate between torque and magnetotropy when analyzing the data. In particular, the example displayed in figure 4 would be an ideal candidate. (Please also include the frequency response data and how the data is processed to obtain fig 4).
- 4) Regarding figure 4: The authors should include in more detail what the graphical elements shown in figure 4 denote. (red highlighted section vs grey highlighted section)
- 5) Figure 5 is not understandable for the broader audience, but apparently important as it demonstrates the capability to measure the phase boundaries. The authors should give a better description of the physical nature of the kink in panel (a) and how they reconstruct panel (b). Also (if possible) they should indicate the phases of the material.
- 6) The magnetotropic effect was initially introduced in the context of directional growth of plants. The authors should indicate the reasons behind the naming of the effect and in addition should give a better hand waving picture for understanding magnetotropy (e.g. when comparing to heat capacity).

Reviewer #3 (Remarks to the Author):

In this manuscript, the authors report on a new experimental method to detect the magnetotropic

coefficient, a second order derivative of the field-angular dependent free energy, of anisotropic quantum materials. They employ a cantilever developed for dynamic mode atomic force microscopy and have succeeded to detect the differential torque of nanocrystalline samples with high sensitivity. They also demonstrated the performance of the apparatus with examples of the measurements of quantum oscillations in the Weyl semimetal NbP and the field-induced transition in RuCl₃. The data presented are of high quality, but they are already known in the literature.

The experimental technique of "resonant torsion magnetometry" reported in the present manuscript is essentially the same with those already used for the magnetometry on nano-size magnets (called "torque differential magnetometry" or "dynamic mode cantilever magnetometry", etc., in the literature). All these measurements make use of the fact that the shift of the resonant frequency of the cantilever, on which a sample is mounted, is proportional to the derivative of the torque (a second derivative of the free energy). Indeed, the expression like eq.(2) in the present manuscript is already given in Ref.20 (Kamra et al, PRB89,184406). Thus, the measurement technique described in the present manuscript is not new.

The novelty of the present work is the application of this technique to anisotropic quantum materials, in particular to a field-induced phase transition that is anisotropic in nature. For this purpose, the authors have derived thermodynamic Ehrenfest relation that relates a discontinuity in the magnetotropic coefficient to other thermodynamic coefficients. In this regard, the authors have made one step further in the application of the method.

That being said, my opinion is that the novelty of the present work is not high enough to warrant publication in Nature Communications.

Reviewers' comments:

Reviewer #1 (Remarks to the Author):

The authors introduce a new thermodynamic quantity the magnetic torque susceptibility or magnetotropic coefficient as a probe for anisotropic materials. They present a theoretical framework, a method for measuring the magnetotropic coefficient, as well as examples of measurements on two selected compounds, NbP and RuCl₃.

To the best of my knowledge, the authors indeed establish a new method for the investigation of (anisotropic) magnetic materials. They successfully outline the major framework of the method. However, I have some hesitations with the manuscript.

1) The manuscript describes a new method to study materials. Traditionally, such manuscripts are published in dedicated journals like Review of Scientific Instruments. The editors should judge whether Nature Communications is a suitable platform. I expect that the present manuscript can make a substantial impact to the field and can stimulate new research and insight. The manuscript focuses on the presentation of the new thermodynamic quantity and method and does not present substantial insight into the physics of a specific material.

We chose the well-studied materials NbP and RuCl₃ to demonstrate and benchmark our technique and highlight the broad impact it will have due to its sensitivity. For example, RuCl₃ is extremely sensitive to interlayer disorder and sliding defects, which have been shown to strongly affect its magnetism. This has led to multiple spurious phase transitions showing up in measurements requiring large sample volumes, such as heat capacity and neutron scattering (see Y. Kubota *et al.* Phys Rev. B 91, 094422). This is a prime example of where intrinsic physics is best uncovered in a microscopic sample. This was achieved in our measurement, which shows a single transition at the AFM phase boundary—representative of a single magnetic domain. Our manuscript identifies an as-yet overlooked link between the magnetotropic coefficient and the specific heat at a second-order phase transition, and thereby provides a new method for extremely sensitive thermodynamic measurements of very small single crystals.

The main reason for a broad audience journal such as Nature Communications over a specialized instrumental journal, such as Review of Scientific Instrument, is the ease of this approach. No apparatus had to be designed, but rather we repurpose the use of a commercial product. While it is a creative approach, we don't believe it is a sufficient technical advance that would be of interest to RSI. However, the broad applicability of a sensitive thermodynamic technique will be of great interest to the condensed matter community interested in correlated materials – where anisotropy plays a key role. In an interdisciplinary journal like Nature Communications, the ease and simplicity of this approach may also attract readers in other fields, such as biophysics or soft matter.

2) Further details are needed in several parts of the manuscript.

a. Equation (1) includes a Taylor expansion in free energy $F(\theta, H)$. The authors should comment on range of validity including quantitative estimates.

We routinely check the lineshape characteristics by scanning frequency through the resonance. We measure at drive voltages up to 100 mV and below this, we do not observe a shift in the resonant frequency. We typically operate at 10 mV, ensuring that the angular deflections are small enough that we are not experiencing higher order corrections to the frequency shift. We have added a figure and discussion with quantitative estimates surrounding this point in the methods section.

b. The authors mention an equilibrium position $\theta_{\tau} = 2\tau/K$. I believe, this should be $\theta_{\tau} = \tau/(K+k)$.

Thank you for pointing this out. We have updated this in the text.

c. The authors briefly discuss and compare torque and magnetotropic coefficient in linear regime. They clearly highlight the benefits of magnetotropic coefficient but remain vague as to limitations arising from non-linear magnetisation. More details and quantitative estimates for non-linear effects on the magnetotropic coefficient are desirable. In fact, the wording of the sentences discussing non-linear effects should be improved.

We naturally look to the linear regime, where the magnetic torque and the magnetotropic coefficient have a well-understood response, to make quantitative comparisons regarding the signal size and sensitivity. Nonlinear magnetic behavior is highlighted in the measurements of the magnetotropic coefficient across the phase boundary in RuCl₃. It is, however, difficult to make quantitative estimates in this regime, as this depends on the physics of the material. We agree that the discussion surrounding nonlinear effects was confusing and that emphasis on the linear magnetic regime was perhaps misplaced. Therefore, we have included discussion of nonlinear magnetization in the last paragraph of page 2 of the main text, as well as a more extended discussion near the end of the revised methods section.

d. The experimental details in the methods sections should be presented more clearly.

We agree. Based on the reviewer comments, and in particular to address the points raised below, we have significantly extended discussion in the methods section. Thank you for the critical feedback. It helped us strengthen the manuscript and make it more broadly accessible. Short answers are included below.

E.g.

i. Not clear if the drive voltage is oscillating

The drive voltage is oscillating. We updated this.

ii. Highlight current-to-voltage converter in sketch

We modified the figure and text in the methods section to discuss this.

iii. Describe role of electrical connection to cantilever (dashed line on picture in Fig6).

Electrical connection is made to the silicon lever via the silver epoxy (now shown in Figure 6 in the methods), presumably to ground the lever for scanning microscopy purposes. We believe this has a negligible effect in our measurement.

iv. The authors indicate that “the electrical resonance of the parallel LC circuit” is measured. This statement should be clarified, inductive and capacitive components should be highlighted in description and sketch.

I assume that the same method as in Ref. 32 is used. If so, this should be clearly referenced.

The shift in the resonant frequency of the cantilever is capacitively-inferred by measuring the impedance across the tuning fork. We fixed our confusing language. We also made changes to the figure and surrounding text in the methods, which we hope clarify the experimental setup.

v. The magnetotropic coefficient is computed from the shift in frequency using equation (2). This requires knowledge of the effective stiffness. The authors provide an equation for the effective stiffness in the methods section $K_0 = 1.63(E I_y / L)$. The authors should provide the numerical value used for K including an error analysis (e.g. stemming from the uncertainty in dimensions and density). In addition, I suggest to clarify whether I_y is assumed independent of position as it was introduced as $I_y(z)$.

In addition, the integral for the energy functional contains $E(z)$. The authors should highlight how the functional is evaluated.

We use the low-field magnetic response (free energy equation in this regime is given below equation 2) and the anisotropic susceptibility in existing literature to calibrate our detected frequency shift. Alternatively, one can estimate the effective spring constant and use equation 2, and in doing so, we find that these quantities are the same within $\sim 10\%$. Most of the uncertainty here comes from the mass of the sample, which is less than 10 ng. We now explain this in detail in the methods section. We also fixed the notation surrounding the energy functional.

vi. An additional frequency shift might occur due to a change in dissipation. The authors should discuss whether this can be neglected.

This is correct. Causality requires that changes in the linewidth will be accompanied by shifts in the resonant frequency. These effects become relevant when the expected shifts in frequency are of order or smaller than the linewidth. Our measured frequency shifts are typically at least 100 times larger than the linewidth. We included a paragraph to this end in the methods section.

In summary, the manuscript presents a valuable step forwards for investigating anisotropic quantum materials as claimed in the title. If the authors can clarify the technical issues raised I am happy to recommend publication and expect that the manuscript will be of broad interest to experimental physicists.

Reviewer #2 (Remarks to the Author):

Dear Editor,

the manuscript by Modic and coworkers describes a technique to measure magnetic phase transitions using a method derived from torque magnetometry. The authors introduce a „magnetotropic“ term into the energy balance of the cantilever and show that they can measure this term by analysis of the resonance frequency. This is of potential great interest when exploring phase boundaries / transitions.

Although I find this method new and interesting I still have various questions regarding the technique, as well as the separation between the torque and the magnetotropic component, before I can recommend the manuscript for publication.

In detail, I encourage to address the following points

1) As shown in Figure 1d, the authors glued a massive object to the cantilever. As shown in the microscope image, the sample seems to exceed the dimensions of the cantilever, warranting the question whether the description of Eq1 holds. Please comment also on the quality factor of the cantilever.

Our use of equation 1 assumes the angular displacement of the lever to be small and the sample to be a rigid object. The resonant frequencies of the crystal are much higher than those of the cantilever. We add a small amount of exchange gas to thermally equilibrate, but don't expect dampening due to this or in the sample attachment to the lever (since the grease used to hold the sample freezes at these temperatures). The only effect of the sample then is to add mass to the lever. To check that we remain in the linear response regime of the lever, we routinely look at the frequency scan characteristics of the resonance for a shift in frequency. Then we operate well below the drive voltages where we see these shifts to ensure that the angular deflections are small enough that we are not experiencing higher order corrections to the frequency shift. We have added a figure and discussion surrounding this point in the methods section.

2) The magnetic field orientation with respect to the crystalline axis of the sample and the cantilever is a bit unclear. A figure / inset would help here.

The first figure was only meant to orient the reader to the principles of the technique. We want to avoid specifics of the crystal for this general technique discussion. We did, however, add the orientation of magnetic field with respect to the lever in figure one as it is important that the magnetic field is oriented in the plane of the lever vibrations. Later in the paper, we clarified crystal orientations w.r.t. the lever when discussing the specifics of the materials.

3) As the authors state, also torque magnetometry can be measured via the frequency shift of the mechanical system. Therefore, I encourage the authors to explain in more detail how they separate between torque and magnetotropy when analyzing the data. In particular, the example displayed in figure 4 would be an ideal candidate. (Please also include the frequency response data and how the data is processed to obtain fig 4).

Thank you for bringing this point to our attention, which was not clear in our previous version. As indicated in Figure 1, the torque leads to a static deflection of the cantilever beam, while the magnetotropic coefficient is responsible for the frequency shift. The present setup detects frequency shifts of the oscillation, and as such is insensitive to the static displacement (as long as it is small enough so that Hooke's law remains valid). The data in figure 4a was taken instead through a conventional torque magnetometry measurement utilizing a piezoresistive cantilever. Like any traditional torque measurements, this technique detects the static deflection angle and thus measures directly the torque. We are very happy that you pointed this out and we made it explicitly clear in the main text and in the figure and the figure caption.

Our measured frequency shift is then directly proportional (according to equation 2) to the data plotted in figure 4b. We use the low-field response and knowledge of the anisotropic susceptibility from the literature to scale the frequency shift into the proper units given in figure 4.

4) Regarding figure 4: The authors should include in more detail what the graphical elements shown in figure 4 denote. (red highlighted section vs grey highlighted section)

We have updated the figure and the figure caption.

5) Figure 5 is not understandable for the broader audience, but apparently important as it demonstrates the capability to measure the phase boundaries. The authors should give a better description of the physical nature of the kink in panel (a) and how they reconstruct panel (b). Also (if possible) they should indicate the phases of the material.

Your comments are very helpful. We hope the current version make things clearer for broad readership. We included the AFM phase in the figure and highlight how we obtain the field-angle position of the transition for each magnetic field. We also rewrote the caption.

6) The magnetotropic effect was initially introduced in the context of directional growth of plants. The authors should indicate the reasons behind the naming of the effect and in addition

should give a better hand waving picture for understanding magnetotropy (e.g. when comparing to heat capacity).

We were not aware of this use of magnetotropic. We have included a couple sentences near the bottom of page 1 explaining why we chose magnetotropic.

Reviewer #3 (Remarks to the Author):

In this manuscript, the authors report on a new experimental method to detect the magnetotropic coefficient, a second order derivative of the field-angular dependent free energy, of anisotropic quantum materials. They employ a cantilever developed for dynamic mode atomic force microscopy and have succeeded to detect the differential torque of nanocrystalline samples with high sensitivity. They also demonstrated the performance of the apparatus with examples of the measurements of quantum oscillations in the Weyl semimetal NbP and the field-induced transition in RuCl₃. The data presented are of high quality, but they are already known in the literature.

The experimental technique of “resonant torsion magnetometry” reported in the present manuscript is essentially the same with those already used for the magnetometry on nano-size magnets (called “torque differential magnetometry” or “dynamic mode cantilever magnetometry”, etc., in the literature). All these measurements make use of the fact that the shift of the resonant frequency of the cantilever, on which a sample is mounted, is proportional to the derivative of the torque (a second derivative of the free energy). Indeed, the expression like eq.(2) in the present manuscript is already given in Ref.20 (Kamra et al, PRB89,184406). Thus, the measurement technique described in the present manuscript is not new.

The novelty of the present work is the application of this technique to anisotropic quantum materials, in particular to a field-induced phase transition that is anisotropic in nature. For this purpose, the authors have derived thermodynamic Ehrenfest relation that relates a discontinuity in the magnetotropic coefficient to other thermodynamic coefficients. In this regard, the authors have made one step further in the application of the method.

That being said, my opinion is that the novelty of the present work is not high enough to warrant publication in Nature Communications.

Thank you for reviewing the manuscript and your interest in our work. The novelty of our manuscript relies on identifying and clearly demonstrating the important thermodynamic aspects of our measurement, which have thus-far been missed. Indeed other groups have used these cantilevers for torque magnetometry, as stated and referenced in our manuscript. The focus of previous studies was the enhanced sensitivity to small changes of torque due to direct

measurements of the derivative $k = d\tau/d\theta$, which is expressed in their naming “torque differential magnetometry”.

In this work, we present a significant step forward in the experimental study of anisotropic quantum materials. Identifying the form of the free energy functional governing a particular system is the root of understanding complex magnetic materials. The Ehrenfest relation demonstrated in this work is a step towards this direction. The entropy associated with a phase transition is one of the most important parameters in this regard, and it presents a critical gap in experiments. Commonly, high quality single crystals of new quantum materials only exist in the form of micro-crystallites. Measurements of the specific heat rely on the precise determination of the heat transmitted into a sample and the associated temperature change. Specific heat measurements are notoriously difficult in very high magnetic fields and at very low temperatures, and they are virtually impossible to do in sub-microgram samples.

On the other hand, our paper reasons that resonant torque magnetometry can access this parameter by a very simple measurement that is standard in high field laboratories, rotating the sample in fixed magnetic fields. We show that measurement of the second-derivative of the free energy provides additional information across a second-order phase boundary that cannot be accessed with torque magnetometry. We also demonstrate the *quantitative* power of this approach by comparing both the specific heat measurements and the specific heat jump inferred by resonant torsion magnetometry. In fact, while the resonant torsion experiment was straightforward, measuring the specific heat of the RuCl₃ crystal was very challenging due to its small mass. Naturally, for the showcasing of the Ehrenfest relations and benchmarking our approach we rely on well-studied test case materials, allowing to contrast our results with existing literature.

We have elaborated this critical point in the manuscript and hope you can recommend our revised manuscript for publication.

REVIEWERS' COMMENTS:

Reviewer #1 (Remarks to the Author):

The authors have added a large amount of additional details and have improved the depth and narrative of the discussion very much. The current manuscript addresses all my concerns about the content of the manuscript raised in a satisfactory manner.

In the previous round, I raised the question of suitability of this manuscript for Nature Communications. I pointed out that the manuscript is a development of a new measurement method including the theoretical framework and the technical details. The manuscript presents very limited novel insight into the physics of the materials studied and rather presents these as benchmarks. These statements remain valid.

The insight into the disorder effects in RuCl₃ represents a marginal step for understanding the physics of RuCl₃ and the authors emphasise that this is merely a demonstration of the strength of resonant torsion magnetometry.

In their rebuttal, the authors mention that the alternative journal, Review of Scientific Instruments, I mentioned as an example would not be appropriate. Whilst this is a separate discussion, I don't agree with this statement. In particular, the authors present a very detailed description of the experimental setup which got even more extended in the revised manuscript. This is exactly what is usually presented in Review of Scientific Instruments. The fact, that a commercial product has been repurposed is not relevant. I see the technical advance exactly in the repurposing.

I agree with the authors, that the method is of broad relevance for the solid state physics community and potentially also for biophysics and soft matter although magnetic anisotropy is less relevant in these areas. Thus, I reiterate that the editors of Nature Communications should decide whether the manuscript fits the scope of the journal. In my view, the manuscript presents the theoretical framework and the technological realisation of a novel technique with considerable interest for the solid state physics community.

Leaving aside the question of matching the scope of Nature Communications, I recommend the manuscript for publication.

Reviewer #2 (Remarks to the Author):

Dear Editor,

the present version of the manuscript is now much clearer and the authors clarified all of my points. Now it is clear, that the authors refer to torque magnetometry solely by considering a static torque and refer to magnetotropy when investigating the frequency response of the cantilever. As pointed out by referee 3, the latter technique is known under several names (torque differential magnetometry, etc.). It is further discussed theoretically in Kamra PRB 89 and experimentally in Kamra EPJB 88 (2015). Therefore, the novelty is utilizing this technique for more complex materials like a Weyl-semimetal and spin-liquid. The present form of the paper is focused onto the technical aspects of „resonant torque magnetometry“ (to cite parts of the title) and therefore I do not see the novelty required for publication in Nature Communications. Therefore, I suggest publishing the work in a more specialized journal.

Reviewer #3 (Remarks to the Author):

The manuscript is improved compared to the original version. I understand that the authors have developed a very useful experimental method that would be of interest for broad audience, and thereby I recommend the manuscript for publication. I would like to ask the authors, however, to check whether the sign of the last term in the free energy (3) is correct. It appears that it is inconsistent with the Maxwell relation given in the subsequent paragraph.

REVIEWERS' COMMENTS:

Reviewer #1 (Remarks to the Author):

The authors have added a large amount of additional details and have improved the depth and narrative of the discussion very much. The current manuscript addresses all my concerns about the content of the manuscript raised in a satisfactory manner.

In the previous round, I raised the question of suitability of this manuscript for Nature Communications. I pointed out that the manuscript is a development of a new measurement method including the theoretical framework and the technical details. The manuscript presents very limited novel insight into the physics of the materials studied and rather presents these as benchmarks. These statements remain valid.

The insight into the disorder effects in RuCl₃ represents a marginal step for understanding the physics of RuCl₃ and the authors emphasise that this is merely a demonstration of the strength of resonant torsion magnetometry.

In their rebuttal, the authors mention that the alternative journal, Review of Scientific Instruments, I mentioned as an example would not be appropriate. Whilst this is a separate discussion, I don't agree with this statement. In particular, the authors present a very detailed description of the experimental setup which got even more extended in the revised manuscript. This is exactly what is usually presented in Review of Scientific Instruments. The fact, that a commercial product has been repurposed is not relevant. I see the technical advance exactly in the repurposing.

I agree with the authors, that the method is of broad relevance for the solid state physics community and potentially also for biophysics and soft matter although magnetic anisotropy is less relevant in these areas. Thus, I reiterate that the editors of Nature Communications should decide whether the manuscript fits the scope of the journal. In my view, the manuscript presents the theoretical framework and the technological realisation of a novel technique with considerable interest for the solid state physics community.

Leaving aside the question of matching the scope of Nature Communications, I recommend the manuscript for publication.

We thank this referee for recommending our manuscript.

Reviewer #2 (Remarks to the Author):

Dear Editor,

the present version of the manuscript is now much clearer and the authors clarified all of my points. Now it is clear, that the authors refer to torque magnetometry solely by considering a static torque and refer to magnetotropy when investigating the frequency response of the cantilever. As pointed out by referee 3, the latter technique is known under several names (torque differential magnetometry, etc.). It is further discussed theoretically in Kamra PRB 89 and experimentally in Kamra EPJB 88 (2015). Therefore, the novelty is utilizing this technique for more complex materials like a Weyl-semimetal and spin-liquid. The present form of the paper is focused onto the technical aspects of „resonant torque magnetometry“ (to cite parts of the title) and therefore I do not see the novelty required for publication in Nature Communications. Therefore, I suggest publishing the work in a more specialized journal.

We thank this referee for the comments they provided that helped us to improve the manuscript.

Reviewer #3 (Remarks to the Author):

The manuscript is improved compared to the original version. I understand that the authors have developed a very useful experimental method that would be of interest for broad audience, and thereby I recommend the manuscript for publication. I would like to ask the authors, however, to check whether the sign of the last term in the free energy (3) is correct. It appears that it is inconsistent with the Maxwell relation given in the subsequent paragraph.

Thank you for recommending our manuscript in Nature Comm. and pointing out the error in our Maxwell relation. We corrected this.